# A Comprehensive Review of *HLA* and Severe Cutaneous Adverse Drug Reactions: Implication for Clinical Pharmacogenomics and Precision Medicine

**DOI:** 10.3390/ph14111077

**Published:** 2021-10-25

**Authors:** Chiraphat Kloypan, Napatrupron Koomdee, Patompong Satapornpong, Therdpong Tempark, Mohitosh Biswas, Chonlaphat Sukasem

**Affiliations:** 1Unit of Excellence in Integrative Molecular Biomedicine, School of Allied Health Sciences, University of Phayao, Phayao 56000, Thailand; chiraphat.kl@up.ac.th; 2Division of Clinical Immunology and Transfusion Science, Department of Medical Technology, School of Allied Health Sciences, University of Phayao, Phayao 56000, Thailand; 3Division of Pharmacogenomics and Personalized Medicine, Department of Pathology, Faculty of Medicine Ramathibodi Hospital, Mahidol University, Bangkok 10400, Thailand; napatruporn.kom@mahidol.ac.th (N.K.); biswas_07pharm@ru.ac.bd (M.B.); 4Laboratory for Pharmacogenomics, Ramathibodi Hospital, Somdech Phra Debaratana Medical Center SDMC, Bangkok 10400, Thailand; 5Division of General Pharmacy Practice, Department of Pharmaceutical Care, College of Pharmacy, Rangsit University, Pathum Thani 12000, Thailand; patompong.s@rsu.ac.th; 6Excellence Pharmacogenomics and Precision Medicine Centre, College of Pharmacy, Rangsit University, Pathum Thani 12000, Thailand; 7Division of Dermatology, Department of Pediatrics, Faculty of Medicine, Chulalongkorn University, Bangkok 10330, Thailand; Therdpong.T@chula.ac.th; 8Department of Pharmacy, University of Rajshahi, Rajshahi 6205, Bangladesh; 9The Thai Severe Cutaneous Adverse Drug Reaction THAI-SCAR Research-Genomics Thailand, Ramathibodi Hospital, Mahidol University, Bangkok 10400, Thailand; 10The Preventive Genomics & Family Check-Up Services Center, Bumrungrad International Hospital, Pharmacogenomics and Precision Medicine Clinic, Bangkok 10110, Thailand; 11MRC Centre for Drug Safety Science, Department of Pharmacology and Therapeutics, Institute of Systems, Molecular and Integrative Biology, University of Liverpool, Liverpool L69 3GL, UK

**Keywords:** human leukocyte antigen, *HLA* genetic polymorphisms, SCARs, pharmacogenomics, precision medicine

## Abstract

Human leukocyte antigen (*HLA*) encoded by the *HLA* gene is an important modulator for immune responses and drug hypersensitivity reactions as well. Genetic polymorphisms of *HLA* vary widely at population level and are responsible for developing severe cutaneous adverse drug reactions (SCARs) such as Stevens–Johnson syndrome (SJS), toxic epidermal necrolysis (TEN), drug reaction with eosinophilia and systemic symptoms (DRESS), maculopapular exanthema (MPE). The associations of different *HLA* alleles with the risk of drug induced SJS/TEN, DRESS and MPE are strongly supportive for clinical considerations. Prescribing guidelines generated by different national and international working groups for translation of *HLA* pharmacogenetics into clinical practice are underway and functional in many countries, including Thailand. Cutting edge genomic technologies may accelerate wider adoption of *HLA* screening in routine clinical settings. There are great opportunities and several challenges as well for effective implementation of *HLA* genotyping globally in routine clinical practice for the prevention of drug induced SCARs substantially, enforcing precision medicine initiatives.

## 1. Introduction

The major histocompatibility complex (MHC) is a group of cell surface proteins that play a pivotal role in T-cell activation because this process mandatorily requires that the T-cell receptor (TCR) engages with the complementary antigenic peptide bound to MHC molecules. In human, MHC is also known as human leukocyte antigen (*HLA*). The *HLA* is the encoded products of the *HLA* gene complex which is located on the short arm of chromosome 6. The *HLA* antigens are classified into two clusters (class I and class II according to their coding gene locus, their function, tissue distribution, and biochemistry. *HLA* class I molecules are encoded by three loci known as classical *HLA*-*A, HLA*-*B,* and *HLA*-*C* genes while *HLA* class II molecules are encoded by classical *HLA*-*DR, HLA*-*DQ, HLA*-*DP* genes. *HLA* class I molecules are expressed by virtually all nucleated cells and present peptides derived from intracellularly expressed proteins to cytotoxic T cells (CD8+) whereas *HLA* class II proteins are typically expressed by professional antigen-presenting cells, such as dendritic cells, and serve to present internalized exogenous protein to T-helper (CD4+) cells. The *HLA* genes are the most polymorphic genetic region in the human genome accounting for the variability in *HLA* molecules expression to present huge variety of peptides. The *HLA* polymorphisms principally influence the shape and electrochemistry of the peptide-binding groove that consequently determine the repertoire of peptides that can bind to a specific *HLA* molecule. The prevalence of specific *HLA* alleles differs significantly among different populations and ethnic groups [1].

The *HLA* system was first recognized and named from experiments in tissue transplantation, as the *HLA* molecules play a crucial role in the adaptive immune response. However, the attention in *HLA* polymorphisms has widened beyond their role as transplantation/transfusion antigens. This is due to *HLA* is also associated with a large number of human diseases in which the immune system is passionately involved such as autoimmunity, cancers or infectious diseases. Additionally, in pharmacogenomics aspect, several drugs can induce immune hypersensitivity responses through interactions with *HLA* molecules, known as adverse drug reactions (ADRs), which is one of the common causes of hospitalization as well as mortality. As *HLA* expression is co-dominant, the susceptibility to drug hypersensitivity depends on the presence or absence of the relevant allele associated to a specific drug [1,2,3,4,5,6,7].

Therefore, the understanding of its inheritance, nomenclature, and application is important for optimal patient care. Moreover, *HLA* genotyping is very importance in solid organ transplantation (SOT), in hematopoietic stem cell transplantation (HSCT), in transfusion practice for platelet refractoriness patient, in the diagnosis of a variety of disease associations and pharmacogenomics applications. The intention here is to provide a brief overview of the most important features of *HLA*, the principle of testing methodologies commonly used in laboratories, and clinical applications of *HLA* with emphasis on pharmacogenomics and precision medicine.

## 2. HLA Molecular Biology

The class I and class II molecules are codominantly expressed but have different tissue distributions. The class I *HLA*, includes the *HLA*-*A, HLA*-*B*, and *HLA*-*C*, are expressed ubiquitously in most nucleated cells but can also be found in platelets, and immature red cells. The class II *HLA*, includes the *HLA*-*DP, HLA*-*DQ* and *HLA*-*DR*, are expressed primarily on professional antigen-presenting cells (i.e., B cells), endothelial cells, monocytes, macrophages, dendritic cells, Langerhans cells as well as activated T cells but can be expressed by most cells including T cells and endothelial cells under inflammatory conditions.

The important biological function of *HLA* molecules is to regulate the activity of the immune system by presenting a complex composed of a self-HLA molecule and the bound nonself peptide or small molecule (i.e., drug) for recognition of clonally expressed “TCR”. *HLA* class I proteins consist of an α chain with a molecular weight of 45 kDa that associates noncovalently with a smaller 12 kDa nonpolymorphic protein, β_2_-microglobulin. On the other hand, *HLA* class II proteins consist of two similar-sized chains of a molecular weight of 33 (α) and 28 (β) kDa associated noncovalently throughout their extracellular portions. Class I molecules present antigens to cytotoxic CD8(+)T lymphocytes, while class II molecules present antigens to helper CD4(+)T lymphocytes. The specificity of this interaction is determined by interactions of the CD4 and CD8 coreceptors with the class II β chain and class I α chain, correspondingly [2,8].

## 3. *HLA* Genomic Organization and Inheritance

The *HLA* genes are closely linked and are located within the most gene-dene and polymorphic region of the human genome also known as “the MHC” on chromosome 6p21.3. The *HLA* class I region contains the genes for *HLA*-*A,* -*B,* and -*C.* The class II region, also known as *HLA*-*D* regions, contains genes for the α and β chains of *HLA*-*DR,* -*DP,* and -*DQ* [7] Figure 1. The close linkage of class I, class II, and class III genes enables the inheritance of the entire set of *HLA*-*A*, -*B*, -*C*, -*DR*, -*DQ*, and -*DP* genes complex from parent to child as one unit that termed as “haplotype”. Genetic crossovers and recombination in the *HLA* region are uncommon. The *HLA* haplotypes conservation results in predictable associations between alleles of *HLA* genes. A specific combination of alleles, either inside class I or class II genes or between class I and class II genes, will occur more frequently than anticipated by chance at the population level. The term “linkage disequilibrium (LD)” is used to describe this phenomena. *HLA* haplotypes may be evaluated using LD, which can provide crucial hints to ensure accurate *HLA* genotype evaluation [9].

## 4. *HLA* Polymorphisms and *HLA* Nomenclature

The *HLA* genes are highly polymorphic in nature which occur as a result of different mutations, meiotic recombination events, and gene conversion events [10]. Majority of the polymorphism found in the class I and II genes occurs in the exons that encode the α-1 and α-2 (class I, exons 2-3)-3 and the α-1 and β-1 (class II, exon 2) domains which is the peptide (antigen binding) cleft [10]. The *HLA* polymorphisms may affect the controlling function of specific immunity and *HLA* allotypes are also involved in unwanted immune reactions such as autoimmune diseases [1], histo-incompatibility as well as drug hypersensitivity, where small therapeutic drugs interact with antigenic peptides to drive T cell responses restricted by host *HLA* genetic polymorphisms [4,7,11,12,13,14]. More than 10,000 *HLA* class I genetic variants and 4500 *HLA* class II-chain genetic variations are produced by polymorphic *HLA* [15,16,17]. Non-covalent interactions between the peptide main chain and conserved cleft residues, as well as sequence-dependent contacts between peptide side chains and polymorphic cleft residues aid peptide binding. The structure and electrochemical environment of the anchor pockets, which are formed by polymorphic *HLA* residues, favor the binding of peptides with complementary side chains at these sites, resulting in an allotype-specific peptide-binding motif [18]. As a result, *HLA* polymorphism has the functional consequence of altering the range of self- and pathogen-derived peptides that can be delivered at the cell surface [18].

The enormous polymorphism of the *HLA* genes necessitates a rigorous naming technique. The WHO Nomenclature Committee for *HLA* System Factors is in charge of identifying new *HLA* genes, allele sequences, and their quality control. As the broad *HLA* antigen groups were initially identified based on their response with antisera in complement mediated micro lymphocytotoxicity tests, the nomenclature for alleles is mostly based on older serological designations. For the current nomenclature, each *HLA* antigens are uniformly named starting with the locus, antigenic specificity, and molecularly typed allele group. *HLA*-prefix designates the MHC gene complex follow by the capital letters indicate a specific gene. Alleles for each of the genes are identified by a four-field series of two- to three-digit numbers separated by a colon. An asterisk “*” sign indicates that typing is performed by a molecular method and the colon “:” is a field separator. For example, *HLA*-*A***02*:*01*:*01*:*01L* is the allele. The 1st field d (*A***02*) refers to a group of alleles that encode for the A2 antigen. 2nd field (*01*) refers to a specific allele, which encodes a unique *HLA* protein (*A***02*:*01)*. This typically describes serologically equivalent proteins that have differential ability to stimulate T lymphocyte responses. The 3rd and 4th numeric fields convey less clinically significant but scientifically important information. The 3rd field refers to a synonymous (silent) mutation in coding region while the 4th field identifies the different alleles in non-coding regions. Some of polymorphisms affect gene expression by creating low-expression (or null) variants. The relevant expression is denoted by the addition of a capital letter at the end of the 4th field [19].

## 5. *HLA* Genotyping: Methods for Identification

The *HLA* testing supports several clinical applications such as transplantation/transfusion, immunogenetics as well as pharmacogenomics. Conventionally, *HLA* antigens were detected using sera with known anti-*HLA* antibodies similar to the methods performed for RBC phenotyping. However, the diversity of *HLA* genes, which are not distinguished by serology, has required employing DNA-based methods for accurate typing. Molecular typing allows for varying levels of resolution when typing the *HLA* allele, depending on the application needs and the amount of time available to complete the test. On the one hand, “low resolution” or “two-digit typing” (example: *HLA*-*A***01*) supplies the first field in the molecular-based nomenclature, which is usually the serologic typing result. For solid organ transplantation, transfusion practice, and certain disease association, low resolution typing is usually sufficient [6]. On the other hand, “high-resolution” or “four-digit typing” (for example, *HLA*-*B***57*:*01*) identifies alleles based on the sequence of the *HLA* molecule’s peptide-binding region, which corresponds to the first two fields of the molecular nomenclature. This level of type does not differentiate between alleles with synonymous alterations or non-coding regions that vary. For bone marrow transplantation, some disease association, and pharmacogenomic testing, high-resolution typing usually necessitates gene sequencing [6]. There is also the intermediate resolution, which contains a subset of alleles that share the digits in their allele name’s first field but excludes certain alleles that share this field.

There are many molecular methods for *HLA* genotyping, for instance, sequence specific oligonucleotide probe hybridization (SSO), sequence-specific primers polymerase chain reaction (SSP), allele-specific polymerase chain reaction (AS-PCR), real-time PCR, and DNA sequence-based typing (SBT). Exons 2 and 3 for *HLA*-*A*, -*B*, and -*C* (class I) genes, and exon 2 for *HLA*-*DR*, -*DQ*, and -*DP* (class II) genes, are the *HLA* polymorphic areas that must be included for typing by these techniques. The comparison of findings to vast lists of potential alleles is required for *HLA* sequence analysis, which generally necessitates interpretive skill and the use of specialist software tools. We provide an overview of the available approaches utilized by laboratories for *HLA* genotyping in this study, outlining the methodological advantages and disadvantages in Table 1. Laboratories must examine a number of criteria when choosing an *HLA* typing technique. The SSP and rSSO take around 4–5 h to complete, making them ideal for clinical circumstances like cadaver kidney transplantation and pharmacogenomics. The SSP is suitable for low to moderate typing volumes, but the rSSO is better suited to big clinical quantities or batch research typing. Due to the use of many primers and probes, SSP and rSSO can reach allele level resolution. For high-resolution typing, however, SBT is the gold standard. Furthermore, serological typing is still helpful for comparing serological results to DNA-based results and determining the existence of null alleles [20,21,22,23,24].

## 6. Severe Cutaneous Adverse Drug Reactions (SCARs)

Severe cutaneous adverse drug reactions (SCARs) are delayed T cell induced hypersensitivity (DTH) including Stevens–Johnson syndrome/toxic epidermal necrolysis (SJS/TEN), drug reaction with eosinophilic and systemic symptoms (DRESS), acute generalized exanthematous pustulosis (AGEP) as described elsewhere [25]. The pictures of different form of SCARs are shown in Figure 2A–C, respectively. These conditions have variation of critical clinical course, skin rashes and severe systemic multi-organ involvement. While other authors have reviewed the associations between *HLA* alleles and SCARs [26,27,28,29,30,31], however, preset review will reemphasize this relationship along with other important factors in this filed.

### 6.1. Stevens–Johnson Syndrome/Toxic Epidermal Necrolysis (SJS/TEN)

SJS and TEN are rare and life-threatening muco-cutaneous hypersensitivity reactions that are in most of the cases drug-induced. Patients’ genetic factors (*HLA* alleles), drug metabolism and interaction of T-cell clonotypes are associated with the pathogenesis [32]. The incidence is ranging from ~2 to 7 per million people per year [33,34], as reported elsewhere. Pediatric SJS/TEN is different from adult SJS/TEN in term of differential diagnosis, etiology, drug exposure, risk of recurrence and outcome [35]. Recurrence is more common in children due to cause of infection [32], relatively lower mortality in pediatric SJS/TEN [33].

There are multiple factors associated with the etiology of SJS/TEN including genetic susceptibilities: *HLA* profiles, individual drug use, drug metabolism, ethnicity-specific association and underlying diseases particularly, hematologic malignancy, HIV infection, liver and kidney diseases. There are some differences in term of the causative drug and associated causes between children and adult. Common culprit drugs in children triggers include sulfonamides, aromatic anticonvulsants (carbamazepine, phenytoin, and phenobarbital), penicillin, and NSAIDs. Other more strongly causative drugs in adults are allopurinol, oxicam, NSAIDs, and nevirapine [34,36,37]. SJS/TEN is considered to be associated with medication when the patient has ingested the suspected agent, usually trigger within 8 weeks prior to the onset of the rash [34]. The typical onset duration is about 4 days to 4 weeks.

Mycoplasma pneumonia infection is the second most common trigger of SJS, particularly in pediatric population and possibly associated with less severe clinical presentation than drug-induced SJS. The other presumptive cause of SJS/TEN from an infection include coxsackie virus, influenza, herpes simplex virus, cytomegalovirus, parvovirus, varicella zoster virus, Epstein–Barr virus, measles virus, human herpes virus types 6 and 7 (HHV-6, 7) streptococcus group A, mycobacterium, and rickettsia [34,38,39] that is noted to occur 1 week prior to the onset of the rash.

The exact immune-histopathology of SJS/TEN is still not fully understood. Drug specific CD8 T-cells and NK cells are the major inducer of keratinocyte apoptosis. Specific T cell receptors recognize a metabolized drug presented by specific *HLA* alleles, followed by the activation of drug-induced cytotoxic T lymphocytes (CTLs) and the release of multiple cytokines, chemokines, signals, and soluble cytotoxic mediators such as Fas-ligand, granulysin, perforin, granzyme B and tumor necrosis factor alpha [40]. IL-15 signal pass through the JAK-STAT pathway, then has downstream to PI3K/AKT/mTOR pathway responsible that effects on NK and CD8 T-cell.

### 6.2. Drug Reaction with Eosinophilia and Systemic Symptom (DRESS)

DRESS is characterized by fever, lymphadenopathy, hematologic abnormalities (eosinophilia, atypical lymphocyte), multi-systemic involvement and viral reactivation. The true incidence of DRESS is still unclear. The younger children seem to be found less than in the adults. Numerous medications have been reported to cause this condition include allopurinol, aromatic/non-aromatic anticonvulsants (carbamazepine, phenobarbital, phenytoin, valproic acid), antimicrobial (ampicillin, cefotaxime, dapsone, ethambutol. isoniazid, trimethoprim-sulfamethoxazole, minocycline, metronidazole), antiviral (abacavir, nevirapine), antihypertensive (amlodipine, captopril), NSAIDs (ibuprofen) [41]. Carbamazepine is the most common causative agents of DRESS in adult [42,43] and pediatric patients [44].

The onset of DRESS symptoms is typically 2–3 weeks from initial exposure of the drug, ranging from 2 to 8 weeks (average 22.2 days, range 0.42–53 days) [42,44,45]. Not only drug specific immune response but also viral reactivation of human herpesvirus (HHV)-6, 7, Epstein–Barr virus (EBV) and cytomegalovirus are associated with this condition. High viral load and antibody titers are the poor prognostic outcome [46]. DRESS is the result of complex interplay of genetic factors (especially *HLA* alleles), immunological response (T-cell), and abnormality of drug detoxification enzymes pathway and herpes virus family member reactivation (HHV-6, 7, EBV, CMV) [41,42,46]. This viral reactivation probably causes of chronic recurrence despite cessation of the culprit drug [46].

Polymorphism in gene encoding drug metabolizing enzyme such as cytochrome P450 (CYP) enzymes, N-acetyltransferase possibly takes part in the pathogenesis of DRESS [47]. Multiple medications including aromatic anticonvulsants are metabolized by the hepatic CYP450 enzymes and oxidation by aromatic hydroxylase may produce the arene oxides which are the excessive toxic metabolites. Alteration of these metabolic enzymes leads to accumulate these toxic metabolites that dysregulated the immune response conducting cell necrosis and/or apoptosis [48]. Toxic metabolite accumulation derived from defect of these metabolized components that alternated the immune response causing cell necrosis and apoptosis. Diagnosis criteria of DRESS is shown in Table 2.

### 6.3. Acute Generalized Exanthematous Pustulosis (AGEP)

AGEP is attributed commonly to drugs, infection and other substance, respectively. It is typically characterized by the acute onset of numerous non-follicular sterile pustules on erythroderma, fever and neutrophilia. The estimate incidence of AGEP is about one to five per million per years [49]. There was variation of patient’s age, whereas mean age range from 40.8–56 years (±21 years) [50,51]. More than 90% of cases associated with medication ingestion including antibiotics (aminopenicillin, sulfonamide, quinolone), anti-fungal (terbinafine), calcium channel blocker and antimalarial agents (hydroxychloroquine) [52,53].

The other causes of AGEP had been described as the contact sensitivity with mercury, lacquer, psoralen combined with ultraviolet A (PUVA) treatment and potent topical NSAIDs [52,53]. AGEP is induced by spider bite, viral infection (Coxsackie B4, cytomegalovirus, parvovirus B19) and bacterial infection (*Mycoplasma pneumoniae*, *Chlamydial pneumoniae*, *Escherichia coli*) and parasitic infestation [52,53]. AGEP commonly occur as a sudden onset, mostly has time interval from drug ingestion within 24–48 h (hour to 25 days) [52,54]. The onset of this condition is variable depend on different drug.

Immune mechanism changes after exposure with the culprit agent forming a drug epitope. Antigen presenting cells present the drug related epitopes by MHC molecules leads to activate specific CD4 and CD8 T cells that reaction as “drug-specific cytotoxic T cells”. CD8 T cells release cytotoxic protein such as perforin, granzyme B and Fas ligand causing apoptosis of keratinocytes within the dermis causing sub-corneal vesicles formation [55].

Furthermore, drug specific CD4 T cells released chemokine (C-X-C motif) ligand 8 (CXCL 8)/IL-8, interferon gamma (IFN-γ), granulocytes/macrophage colony stimulating factor (GM-CSF). CXCL 8/IL-8 is a potent neutrophilic chemotactic chemokine the recruited neutrophils. GM-CSF protects these recruited neutrophils from apoptosis, whereas IFN-γ synergistic produces of CXCL 8/IL-8 from surrounding keratinocytes. Neutrophil infiltrations mediate transform of sub-corneal vesicles to sterile pustules [52,56].

CD4 T cells occasionally stimulate Th2 cytokine pattern to produce IL-4, IL-5, a potent stimulator of eosinophilic differentiation. This uncommon event is the cause of eosinophilia that had been found about 30% of AGEP patients [53]. Th-17 cells, IL-17, IL-22 have been proposed to be pathogenesis of AGEP. The synergist effects of IL-17 and IL-22 expression by neutrophils, mast cell and macrophage are inducible CXCL 8/IL-8 production from keratinocyte [55]. Furthermore, the increasing of IL-17 level in subcorneal pustules of AGEP patients can be detected [57]. The recent studies about genetic predisposing identified mutation IL-36RN in AGEP patients [58]. These mutations lead to increase expression of various pro-inflammatory cytokines and chemokines such as IL-36 signaling, up-regulating IL-1, IL-6, CXCL 8/IL-8 production and neutrophil recruitment [54,59]. Etiology, pathology and treatments of SCARs have been extensively described as reviewed by other authors [60,61,62].

## 7. Potential Mechanism of *HLA*-Associated Drug Hypersensitivity

The mechanisms behind drug-induced delayed hypersensitivity reactions (DHS) may be explained by three different theories: hapten/prohapten theory, pharmacological interaction theory (p-i), and repertoire alteration theory (Figure 3).

### 7.1. Hapten/Pro-Hapten Theory

The ‘Hapten concept’ or ‘Haptenisation’ was first introduced by Landsteiner and Jacob in 1936. As per this model, the hapten, a substance that has <1000 Dalton molecular weight capable of triggering the immune response when it binds with larger protein or peptides. The electrophilic or reactive chemicals or drugs behave as hapten and initiate the immune response. However, the hapten hypothesis is modified into ‘prohapten theory’ as many inert substances are reported with DHS. According to this concept, the toxic metabolites of drugs can bind with protein and become immunogenic (Figure 3a). Therefore, the non-reactive drug acquires reactivity by the virtue of its metabolites. For instance, sulfamethoxazole is pro-haptane which is converted into a proactive hydroxylamine metabolite by CYP2C9 at first then into the unstable and readily protein binding nitroso sulfamethoxazole and this pro-haptane complex can initiate the immunological response as it is processed by an antigen-presenting cell (APC) and presented to T cell through MHC or *HLA* [63]. The triggered immune responses could be humoral that is anaphylaxis or cell-mediated response, the delayed type of T-cell mediated immune reactions.

According to this theory, the non-reactive drug binds directly to the TCR or *HLA* by a non-covalent bond. The T cell activation requires the interaction between drugs and *HLA* or TCR or both and usually, the degree of interaction is metabolism independent. The interaction between drug and TCR/*HLA* may not always end up T-cell response. Likewise, the drug may inhibit another structurally similar drug’s interaction with TCR/*HLA*. A typical example is that the sulfanilamide suppresses sulfamethoxazole induced proliferation [64].

### 7.2. The Pharmacological-Interaction (P-I) Theory

The P-I theory can be classified into two, the p-i TCR interaction in which a drug binds to TCR and activates T cells via strengthening its interaction with peptide-*HLA* (pHLA) [65]. The drug may bind to TCR itself with high affinity at sites that contact with pMHC resulting in T-cell activation, even in the presence of different peptides or allogeneic *HLA* [64,66,67]. In case of carbamazepine-induced hypersensitivity, the carbamazepine interacts with both TCR and *HLA*-*B***15*:*02*, suggesting that it exhibits the features of both p-i TCR direct p-i and p-i *HLA* indirect p-i [68] as shown in Figure 3b.

### 7.3. Repertoire Alteration Theory

In this model, the drug binds with MHC binding site by non-covalent bond and altering the chemistry of binding cleft and endogenous peptide repertoire. This alteration modifies the selection and presentation of peptide ligands necessary for TCR activation. Studies show that abacavir is found to bind non-covalently to *HLA*-*B***57*:*01* molecule, causing changes in the peptides binding ability of *HLA*-*B***57*:*01* molecule causing an alteration in the repertoire of endogenous peptides presented to TCR (Figure 3c).

## 8. *HLA* and CD8+T-Cells Provide the Immunogenetics Basis of Systemic Drug Hypersensitivity

MHC is unique for every individual and every tissue. All nucleated cells have MHC-I, MHC-II complex present in APCs such as dendritic cells, macrophages, and neutrophils. All nucleated cells contain the class I MHC molecule (*HLA*-A, *HLA*-B, and *HLA*-C) as transmembrane glycoproteins on their surface, they are encoded by genes presented at *HLA A*, *HLA B*, *HLA C* loci. T cells that express CD8+ molecules react with this class I MHC molecules, so all the infected cells can act as APC for these CD8+T cells. Class II MHC molecules are presented only in regular APC such as dendritic cells, B cells and macrophages, and so on. These molecules are encoded by genes that are located in the *HLA*-*DP*, -*DQ*, or -*DR* loci, and T cell expressed with CD4+ reacts with them. The Source of allergen is usually from the cytoplasm (viral infection) in CD8+T whereas in CD4+T cells antigen is from extracellular. Cytokines are produced by both CD8+T cells and CD4+T cells, hence both of them are capable to produce DTH.

Antigen processing type is the important determinant of T-cell response. If antigen comes from outside, MHC II molecule present this to T cell which is expressed with CD4+ to produce a response, then the differentiated CD4+T cell, produce Helper T cell (TH1) which can stimulate cytokinins production through IL2 and IFN gamma, on the other hand, TH2 produce antigen-specific antibodies by IL4, IL 5 and IL 10. CD8+T cells follow the same pattern in producing cytokinins and IFN gamma which can initiate DTH through TC1 and TC2.

Studies have confirmed that the histology of drug-induced eruptions and alloimmune reactions are similar, because in both cases the predominant infiltration cells are CD8+T cells, in contrast, CD4+T cells are presented deeper dermis in both conditions. Another mechanism of CD8+T cells and drug hypersensitivity is that some drugs are transformed into chemically reactive haptens, which can bind with any peptides and can be considered as drug antigen. Keratinocytes contain CYP450 enzymes that can do this transformation. For example, sulfanilamide and phenytoin are metabolized into their reactive metabolites of aryl-hydroxide and epoxide respectively, and usually, these reactive metabolites are detoxified by enzymes, but if there is any genetic variation in genes that encode these enzymes may initiate the immune reaction by endogenous pathway and presented to CD8+T cells.

In another proposed mechanism, drugs can directly bind (non-covalently) with existing peptide/MHC class I or class II complexes and presented to CD4+ and CD8+T-cell clones, this happens even in the absence of metabolism and processing. In drug-induced bullous eruptions, CD8+T cells play a major role by initiating the cytotoxic effect by producing potent medicated perforin, So the study has suggested that the sulfonamides induced cutaneous eruption in AIDS patients is mainly mediated by CD8+class I-restricted, perforin producing, cytotoxic DTH effector T cells.

## 9. Pharmacogenomics and *HLA* Research across the World

Polymorphism of classical *HLA*-*A*, *HLA*-*B*, *HLA*-*C*, *HLA*-*DR*, and *HLA*-*DQ* genes varies among populations in the respect of the frequencies of alleles and haplotypes particular to population groups. Ethnic-specific genetic variation information is vital for clinical implementation, especially, to identification of good pharmacogenetic markers. There have been reported associations between various *HLA* alleles and different adverse drug reactions, as described in Table 3 and Table 4.

## 10. Drug Hypersensitivity and *HLA* Alleles: Translational Research into Clinical Practices

### 10.1. Association of Abacavir Hypersensitivity with HLA-B*57:01

Fever, rash, malaise, nausea, vomiting and diarrhea are all symptoms of abacavir hypersensitivity syndrome, which can be lethal if the drug is reintroduced. About ten years ago, the initial discovery that it was linked to the *HLA*-*B***57*:*01* allele was an encouraging development for drug hypersensitivity research [91]. It provides a unique model to explore the pathophysiology of drug hypersensitivity because of its narrow *HLA* limitation and high PPV of 47.9% Martin et al. [92] proposed that the innate immune system is involved in abacavir hypersensitivity reactions, in which abacavir stimulates APC through the HSP70-mediated Toll-like receptor pathway. However, there have been no additional complaints to back up this claim. The activation of abacavir-specific T cells requires the *HLA*-*B***57*:*01* molecule, as established by the creation of abacavir-specific T-cell lines from abacavir-naive *HLA*-*B***57*:*01* positive donors. The CD8 T cell-driven, drug-specific response was abrogated if *HLA*-*B***57*:*01* was replaced by *HLA*-*B***57*:*03* or even a single, essential amino acid alteration. This suggests that a single amino acid in *HLA*-*B***57*:*01* could be the deciding factor in abacavir hypersensitivity development.

Over a decade, the *HLA* genetic variability has been identified as the major predisposing risk factor for SCARs in which the association between the *HLA*-*B***57*:*01* allele and abacavir hypersensitivity syndrome (AHS) is one of the most confounding evidence. To date, several important studies revealed that approximately 2–8% of patients are experiencing AHS warranting discontinuation of abacavir therapy. Consequentially, inheriting *HLA*-*B***57*:*01* allele shows a highly specific association with the AHS that has been validated in patients taking abacavir in numerous clinical studies [93,94,95,96,97].

Due to strong association between the *HLA*-*B***5701* allele and AHS, several studies have demonstrated that genetic pre-emptive screening of *HLA*-*B***5701* before starting abacavir is extremely useful and is reducing AHS substantially [95,96,98,99]. Pre-emptive screening of *HLA*-*B***57*:*01* positive patients treating with alternative antiretrovirals and starting of abacavir only in *HLA*-*B***57*:*01* negative patients have eradicated incidences of laboratory confirmed AHS in significant number of the patients [94,100]. This has geared relevant international drug expert groups including pharmacogenomics (PGx) expertise to generate guidelines for suggesting pre-emptive *HLA*-*B***57*:*01* genetic testing to advise abacavir prescription accordingly as provided by the Clinical Pharmacogenetics Implementation Consortium (CPIC) and the Infectious Diseases Society of America (IDSA). Additionally, both the European Medicines Agency (EMA) and the US Food and Drug Administration (USFDA) have concordantly included boxed warnings on the drug label, which advocate in advance *HLA*-*B***57*:*01* screening while taking abacavir and that recommend *HLA*-*B***57*:*01* positive patients should not be taking abacavir [97,101,102,103,104].

The prevalence of *HLA*-*B***57*:*01* allele varies in different ethnicities profoundly. A recent study after accumulation of data from major ethnic groups revealed that the prevalence of *HLA*-*B***57*:*01* allele was highest in South Asia (227.4 million allele carriers; carrier rate 9.3%) followed by Europe (44.4 million allele carriers; carrier rate 6%). In contrast, the prevalence of *HLA*-*B***57*:*01* allele was considerably lower in East Asia (16.2 million allele carriers; carrier rate 1%) [97]. In Thailand, the prevalence of *HLA*-*B***5701* allele is ~2–4% [17,97].

The screening of *HLA*-*B***57*:*01* allele has increased steadily since its first inclusion in standard of care which was accompanied by a decreasing incidence of definite or probable AHS over the decade. As a considerable proportion of patients are not screened in many parts of the world for *HLA*-*B***57*:*01* allele while taking abacavir, certain proportion of abacavir induced hypersensitivity reactions are remained still though it is preventable [96]. A recent cost-effective analysis showed that pre-emptive testing of *HLA*-*B***57*:*01* allele was cost-effective in most of the countries while taking abacavir [97]. Countries where *HLA*-*B***57*:*01* testing is still not on the standard care, appropriate strategy and planning including preparing national guidelines is needed to launch such testing program to prevent abacavir driven hypersensitivity reactions.

### 10.2. Association of Carbamazepine Hypersensitivity with HLA-B*15:02

In Asian populations, *HLA*-*B***15*:*02* allele is highly linked to carbamazepine (CBZ)-induced SJS/TEN. The *HLA*-*B***15*:*02* testing before CBZ prescription has been shown to prevent CBZ-SJS/TEN. However, there have been reports of patients who acquired CBZ-SJS despite not having *HLAB***15*:*02.* They describe a Thai patient who developed CBZ-SJS despite being negative for *HLA*-*B***15*:*02*, and who was later found to have *HLA*-*B***15*:*21*, an *HLA*-*B75* serotype marker that is identical to *HLA*-*B***15*:*02*, *B***15*:*11*, and *B***15*:*08*.

They hypothesized that if all *HLA*-*B***15*:*02* carriers were barred from receiving CBZ, another prevalent *HLA*-*B75* serotype marker, particularly *HLA*-*B***15*:*21* would affect the development of CBZ-SJS. To test this hypothesis, we reviewed published association studies in Asian populations excluding Japanese and Korean studies, which have been found to have no association with *HLA*-*B***15*:*02* pooled genotype data, excluded all cases and controls with *HLA*-*B***15*:*02*, and then looked at the association between *HLA*-*B75* serotype markers and CBZ-SJS. A tertiary structure of the protein component is required for comprehensive examination because the serotype manifests in a specific protein form. They not only built a tertiary structure in this study, but they also ran an in-silico analysis to compare all *HLA*-*B75* structures and the molecular interaction between the CBZ molecule and all *HLA*-*B75* serotype molecules to the *HLA*-*B***15*:*01* serotype. While the fact that *HLA*-*B***15*:*02* screening is successful in preventing CBZ-SJS, a handful of persons have developed CBZ-SJS despite having neither *HLA*-*B***15*:*02* nor the two known related markers, *HLA*-*A***31*:*01* and *B***15*:*11*.

Because *HLA* allele frequencies differ widely amongst groups, the statistical significance of association studies involving *HLA*-*B* allele frequencies has likewise varied. *HLA*-*B***15*:*21*, for example, is common in Southeast Asian and neighboring populations with a high *HLA*-*B***15*:*02* frequency. In real-world association studies, however, only *HLA*-*B***15*:*02* and *B***15*:*11* have been demonstrated to be related with CBZ-SJS. The first positive connection between CBZ-SJS and *HLA*-*B***15*:*21*, but not *HLA*-*B***15*:*08*, the *HLA*-*B75* serotype marker with very low frequencies in Asian populations, is presented in this study. If an unmistakable link exists between all *HLA*-*B75* serotype members and CBZ-SJS, a screening policy for all *HLA*-*B75* serotypes should be developed in order to maximize the advantage of *HLA*-*B* screening prior to CBZ prescription [105,106,107,108].

Although strong association of the *HLA*-*B***15*:*02* allele with CBZ-induced SJS/TEN has well established in numerous clinical studies and also supported by the meta-analyses [105,109,110], however, CBZ-induced SCARs associated with other *HLA*-*B* variants has not well-noticed yet [105]. This is because, a recent in silico study showed that a Thai female patient negative for *HLA*-*B*:*15*:*02* treated with CBZ was developed SJS due to the presence of the *HLA*-*B*:*15*:*21* allele [108]. A recent study conducted by Sukasem C et al. 2018 found that CBZ-induced SJS/TEN was significantly associated with *HLA*-*B***15*:*21* allele compared with CBZ-tolerant controls (OR = 9.54; 95% CI 1.61–56.57; *p* = 0.013). This is also in line with the findings identified in other clinical studies [105]. Additionally, some studies established the associations of other *HLA*-*B75* serotype, i.e., *HLA*-*B*:*15*:*08* and *HLA*-*B***15*:*11* with CBZ induced SJS/TEN [111,112,113,114,115]. The findings of these studies urge that not only just *HLA*-*B***15*:*02* allele but also *HLA*-*B75* serotype, i.e., *HLA*-*B*:*15*:*02*, *HLA*-*B***15*:*08*; *HLA*-*B***15*:*11* and *HLA*-*B***15*:*21* should take into considerations for optimizing safety of CBZ. A panel of PGx biomarkers consisting of these *HLA*-*B75* serotype should include in future screening policy to cover all potential risk alleles to prevent CBZ-induced SCARs substantially.

The biggest challenge for finding a suitable alternative drug for patients positive with *HLA*-*B75* serotype especially for *HLA*-*B***15*:*02* positive patients would slow down the rapid growth of precision medicine (PM). This is because, some doctors prefer to prescribe oxcarbazepine (OXC) instead of CBZ in *HLA*-*B***15*:*02* positive patients but this should not be the ideal option as OXC is also strongly associated with SJS/TEN in *HLA*-*B***15*:*02* positive patients [105,116]. Other aromatic anticonvulsant drugs, e.g., phenytoin, phenobarbital or lamotrigine would be he preferable choice although need careful clinical monitoring, as some of these drugs may also weakly associated with SCARs due to either by the *HLA*-*B***15*:*02* allele or by other genetic variants, e.g., *CYP2C9* [117,118,119,120].

Along with *HLA*-*B75* serotype, *HLA*-*A* genetic variability also need consideration for optimizing safety of CBZ. For example, the *HLA*-*A***31*:*01* allele was significantly associated with CBZ-induced DRESS but not for CBZ-induced SJS/TEN in Chinese and European patients [121,122]. A very recent study conducted by Sukasem C et al. in 2020 found a strong association of *HLA*-*A***33*:*03* with DRESS but no association of *HLA*-*B***15*:*02* allele with CBZ-induced DRESS in Thai patients [123].

From colligating overall evidence, it is concluded that *HLA*-*B***15*:*02* allele is a phenotype specific biomarker related to CBZ-induced SJS/TEN but not other SCARs, i.e., maculopapular exanthema (MPE), DRESS. Additionally, both *HLA*-*A* and *HLA*-*B75* serotype need important considerations for reducing SCARs in patients taking CBZ.

Genetic variability of *HLA* gene at population level in different ethnic groups varied widely which is accountable for different degree of drug hypersensitivity reactions as observed in these diverse populations [37,123,124,125,126,127,128]. Therefore, pharmacogenetics biomarkers for preventing SCARs may vary accordingly from one country to another country. For example, due to high prevalence of *HLA*-*B***15*:*02* allele and strong association of this allele with CBZ-induced SCARs in Southeast Asia, Chinese and Taiwanese, the FDA and CPIC recommend to screen *HLA*-*B***15*:*02* allele in these population before initiation of CBZ therapy. In contrast, because of high prevalence of *HLA*-*A***31*:*01* allele and strong association of this allele with CBZ-induced SCARs in European, Japanese and Korean population, the FDA and CPIC recommend to screen *HLA*-*A*:*31*:*01* allele in these population before starting CBZ to prevent SCARs magnificently [106,123,129]. These ethnic specific recommendations are also supported from a recent analysis showing that pre-emptive genotyping of *HLA*-*B***15*:*02* was cost-effective in East and South Asian populations only but *HLA*-*A*:*31*:*01* testing was likely to be cost-effective almost globally while taking CBZ [97]. Although such differences at populations level may pose a profound challenge to prevent drug hypersensitivity reactions especially when PGx cannot be uniformly and selectively translated into clinical practice, however, a population-wide approach for inventing selective PGx biomarkers may overcome this limitation [123].

### 10.3. Association of Allopurinol Hypersensitivity with HLA-B*58:01

Allopurinol-induced SJS/TEN is significantly linked to the *HLA*-*B***58*:*01* allele. While Hung et al. did not distinguish allopurinol-induced SJS/TEN from DRESS, more than half of the patients in that study had DRESS, and all of them had the *HLA*-*B***58*:*01* allele, indicating that allopurinol induced DRESS is extremely likely to be related with *HLA*-*B***58*:*0*. In a Korean investigation, this substantial link with allopurinol-induced DRESS was also verified. In contrast, a recent Australian study discovered that none of the 12 patients with allopurinol-induced MPE had *HLA*-*B***58*:*01*, possibly implying that this link is only observed in more severe phenotypes [130]. Furthermore, a Korean study of allopurinol-treated patients with chronic renal insufficiency found that 18% (9/50) of *HLA*-*B***58*:*01* positive patients had allopurinol-induced SCARs [131,132]. This is substantially greater than the previously predicted *HLA*-*B***58*:*01* PPV of 2.7 percent. In addition, a recent study indicated that a daily intake of 200 mg or more was linked to a higher risk of SJS/TEN [133]. Combining these findings, it is possible that either renal impairment or greater doses may result in increased serum levels of allopurinol and/or its metabolite, oxypurinol, which raises the risk of drug-specific T cell generation. In addition to the availability of drug levels and *HLA* associations, viral infections have long been known to play a role in drug hypersensitivity [91].

Unlike *HLA*-*B***15*:*02*, the *HLA*-*B***58*:*01* allele is not an ethnic specific biomarker rather it is a universal biomarker, as the prevalence of *HLA*-*B***58*:*01* allele is widely distributed across the globe [97,134,135,136,137]. Additionally, *HLA*-*B***58*:*01* allele is not a phenotype specific biomarker, as the association of *HLA*-*B***58*:*01* with allopurinol induced SJS/TEN/DRESS/MPE has been established in numerous clinical studies [130,138,139,140].

A recent guideline released from the American College of Rheumatology (ACR) for the management of gout has recommended conditional *HLA*-*B***58*:*01* allele testing before starting allopurinol therapy for the patients of Southeast Asian descent (e.g., Han Chinese, Korean, Thai) and African American due to having higher prevalence of *HLA*-*B***58*:*01* allele in these ethnic groups although the certainty of evidence for this recommendation was very low [141]. The ACR also strongly recommended to start allopurinol in daily doses of ≥100 mg in general patients but in particular, even lower doses in patients with chronic kidney disease (CKD) because high plasma allopurinol concentration in CKD patients may be susceptible to the greater risk of SCARs as evidenced in multiple studies [141,142,143,144,145,146,147]. Additionally, special attention is required when prescribing allopurinol to elderly patients, as age related degradation of renal function in these patients may further exacerbated the risk of developing SCARs [142,143]. All these factors as discussed above need important clinical considerations when preparing national guidelines for implementing PGx of allopurinol in clinical practice.

## 11. International Clinical Recommendations for *HLA* Genotyping

International pharmacogenetic working group such as the CPIC, the Dutch Pharmacogenetics Working Group (DPWG) and the Canadian Pharmacogenomics Network for Drug Safety (CPNDS) are reviewing the evidence of drug-gene interactions and are developing PGx based dosing guidelines for more than a decade [148,149,150,151].

Each of these PGx working group has its own method of assessing drug-gene interactions (DGIs), e.g., some group consider one drug and one gene for developing guidelines whereas others include one or more drugs with two to three genes. In addition, strategy of grading scientific evidence and the strength of the recommendation differ among these PGx-based dosing guideline developers [148,152,153,154,155]. Of note, only the DPWG and CPNDS provide PGx testing recommendations regarding a specific DGI for implementation in daily clinical practice.

In general, PGx-based dosing guideline development process consists of these steps (i) a systematic literature search of the DGI of interest; (ii) critical assessment of the level of evidence for this particular DGI (iii) Based on robust evidence, clinical practice recommendations are prepared by the PGx expert members and presented in a relevant workshop meeting; (iv) PGx-guideline expert members then undergo internal review of draft guidelines based on the feedback received from the workshop meeting; (v) At last, the practice guideline is reviewed externally by relevant expertise and members of the intended target audience as well to finalize the guideline [148,153].

To date, The CPIC has assessed more than 400 DGI pairs and has published PGx-based dosing recommendations on ~106 DGI pairs with sufficient evidence in which prescribing action is required for at least 24 published guidelines. The CPNDS has reviewed considerable number of DGI pairs and provide dosing recommendations on at least 13 DGI pairs. The DPWG also reviewed more than 100 DGI pairs and published recommendations on considerable number of DGI pairs in which at least 60 DGI pairs are required action such as either dose adjustment or monitoring toxic effects [148,155]. The total number of drugs affected by the *HLA* genetic variability have been reviewed fully or partly by all these PGx working groups and provided guidelines accordingly.

As shown in Table 5, there were strong associations between certain *HLA* genetic variants and drug hypersensitivity/SCARs, as discussed before, and several international PGx working groups, e.g., CPIC, DPWG, CPNDS and medicine regulatory bodies such as US-FDA have recommended *HLA* genotyping for a number of drugs affected by the *HLA* genetic variants to optimize the safety of these medications.

The evidence based PGx-guided dosing recommendations are accelerating clinical decision taken by the healthcare professionals for implementing these recommendations into routine clinical practice. Although there are minor inconsistences, however, these guidelines have great value in terms of strong evidence and unique profiles and may therefore consider as ‘gold standard’ for many countries to assess drug response variability due to PGx activity and to generate national drug policy as well.

## 12. Approach to the *HLA* Genotype Screening in Clinical Implementation

Generally, the prevalence rates of *HLA* genetic polymorphisms may be considered as reference to decide which patients should be screened pre-emptively or reactively. For example, the FDA label states that in Thailand, Malaysia, Hong Kong and Philippines, over 15% of these population is *HLA*-*B***15*:*02* positive compared to ~10% in Taiwan and ~4% in China. However, the prevalence of *HLA*-*B***15*:*02* ranges from ~2–4% in some parts of India, but in Korea and Japan it is prevalent in less than 1% of the population. In contrast, the *HLA*-*B***15*:*02* allele is largely absent in not Asian ethnic groups such as Caucasians, African-Americans and Hispanics [129,156]. As over 90% of patients treated with CBZ may be at risk of developing SCARs, e.g., SJS/TEN within the first few months of treatment, therefore, this information is emphasizing the need for genetic screening of patients taking CBZ being at risk of experiencing SCARs [156].

Genetic screening of *HLA*-*B***15*:*02* in Asian patients taking CBZ demonstrated that substantial number of CBZ-induced SCARs were prevented due to this genetic screening and suggesting that, the simple and rapid *HLA* screening tests in the routine clinical setting are very important for widespread uptake of population screening to prevent morbidity and mortality associated with SCARs [105,121,137,157,158,159]. As discussed above, not only just *HLA*-*B***15*:*02* allele but the *HLA*-*B75* serotype, i.e., *HLA*-*B***15*:*02, HLA*-*B***15*:*08, HLA*-*B***15*:*11* and *HLA*-*B***15*:*21* as well as the *HLA*-*A***31*:*01* and *HLA*-*A***33*:*03* were associated with significantly increased risk of SCARs for the patients taking CBZ. For preventing SCARs, screening of *HLA* genotype may be implemented by the following model as shown in Figure 4 to adhere in routine clinical practice.

## 13. Reimbursement and Cost Effectiveness Screening of *HLA*

Cost-effectiveness of genetic testing is often an important part of debate especially in the case of reimbursement issues. If the genetic screening is undertaken based on the recommendations of high-quality recommending body, e.g., the USFDA or CPIC, there may be high chance to get reimbursement from their own country after claiming the cost of screening especially when the patients are taking treatment from overseas, e.g., from Thailand. This may facilitate rapid and reasonably large uptake of genetic screening of overseas patients in the countries where such facilities are available, e.g., in Thailand. 

Although it is evident that PGx consideration is optimizing the safety or efficacy of many clinically important medications and is reducing the morbidity and mortality, however, genetic testing for the integration of PGx into routine clinical practice must be cost-effective [117,156,158]. Cost-effectiveness analysis of *HLA* screening to prevent SCARs compare to treatment costs without *HLA* screening has been undertaken in some studies. Using a computational model, Tiamkao and colleagues showed that the total cost savings would be approximately USD 3285 in 100 cases if the *HLA*-*B***15*:*02* genotype screening was performed prior to treatment with CBZ [158,160]. Another study conducted by Dong et al. examined detailed cost-effectiveness analysis of *HLA*-*B***15*:02 screening in newly diagnosed epilepsy patients in Singapore. The findings of this study showed that genetic screening was more cost-effective in Singaporean Chinese and Malaysian populations where the prevalence of *HLA*-*B***15*:*02* was more than 5% compared to Singaporean Indians where the prevalence of *HLA*-*B***15*:*02* was less than 2.5%. This study concluded that screening was likely to be cost-effective where the prevalence of *HLA*-*B***15*:*02* was at least greater than 2.5% [158,161].

Cost-effective analysis was also carried out for *HLA*-*B***58*:*01* in patients taking allopurinol. In a Thai study lead by Saokaew et al. showed that pre-emptive *HLA*-*B***58*:*01* screening was cost-effective before initiating allopurinol therapy after calculating all associated direct costs of alternative treatment, genetic testing, SCARs, quality adjusted life years (QALYs) gained and gout management [158,162]. This is also in line with the findings of a Korean study where the investigators concluded that pre-emptive screening of *HLA* prior to treating gout with allopurinol was cost-effective [158,163].

The success of *HLA* screening and cost-effectivity will be dependent on some important considerations. Firstly, the availability of alternative medications suggested due to *HLA* positivity, and it should be less expensive. Secondly, the prevalence of risk alleles associated with SCARs should be reasonably high, at least greater than 2.5% in the screening population. Lastly, the cost of molecular *HLA* genotyping at the clinical setting should be minimum to ensure universal uptake of screening programs in the risk populations [158].

## 14. Drug Hypersensitivity and *HLA*: Clinical Implementation in Thailand

### 14.1. Genomics Thailand

Enforcement of PM initiatives by the president of USA in 2015 has facilitated the government of Thailand to launch the Genomics Thailand Initiative (GTI). Thai government has established GTI in 2019 in order to create a genomic database of 50,000 Thai people to advance PM and PGx research in Thailand. Robust evidence found as part of GTI will then help to develop PGx-based dosing guidelines and will serve as focal point for regulatory framework [117,164]. The GTI is working along with Thailand’s Center for Medical Genomics (CMG) and Center of Excellence for Life Sciences (TCELS) as a collaborative research team for bridging the gap between PM and clinical practice. The outcome of GTI project may benefit the country’s economy by 70 billion baht annually either by intervening the five major diseases (i.e., stroke, ischemic heart disease, diabetes, cancer and HIV infection) or by interfering the cost of PGx test, treatments etc. The PGx research undertaken at Mahidol University, Bangkok, Thailand suggested that approximately six billion baht (~USD 185 million) was probably saved due to only undertaking PGx screening test from 2013 to 2018 [164]. Recently, the GTI has set a 20-year roadmap outlining various objectives. One of the very important objectives of the GTI is to establish robust PGx evidence for at least 14 therapeutic areas (oncology, respiratory, infectious diseases, endocrinology and metabolism, gastroenterology and hepatics, rheumatology, anesthesiology, chronic kidney diseases, hematology, transplantation, neurology and psychiatry, adverse drug reactions: SCAR and DILI, cardiovascular diseases, cannabinoid compound) which may accelerate the translation of PGx into routine clinical practice in the form of PM [117]. The outcomes of GTI will profoundly enhance the development of updated national health policy in Thailand.

### 14.2. From Research into the National Policy: A Model of HLA-B*15:02

#### 14.2.1. Carbamazepine

Through funding from the TCELS, PGx research started in June 2004 in Thailand [117,165]. Among many PGx research projects, CBZ-induced SCARs had been considered for investigation because of the *HLA* genetic variability affecting hypersensitivity of CBZ [166]. A retrospective study conducted by Locharernkul et al. in 2008 first documented the association of CBZ-induced SJS/TEN with *HLA*-*B***15*:*02* allele in the Thai population [167]. Another large retrospective case-control study lead by Tassaneeyakul et al. in 2010 revealed a strong association between the *HLA*-*B***15*:*02* allele and CBZ-induced SJS/TEN in Thai population [168]. Further study was conducted in 2012 by Kulkantrakon K et al. and reported the strong association between *HLA*-*B***15*:*02* allele and CBZ-induced SJS/TEN. In 2013, two studies assessed the cost-effectiveness of *HLA*-*B***15*:*02* allele screening in Thailand and concluded that pre-emptive screening of *HLA*-*B***15*:*02* allele was cost effective before prescribing CBZ after calculating all other associated costs. These findings expedite the health policy maker of Thailand to take initiatives for starting PGx testing of *HLA*-*B***15*:*02* allele prior to CBZ therapy, and therefore, it was commenced from the 1 October 2013 as part of the standard care in Thailand to prevent CBZ-induced SJS/TEN [165].

Later, based on the robust evidence of the association of *HLA*-*B***15*:*02* with CBZ-induced SJS/TEN from some other prospective studies, Epilepsy Society of Thailand published national guideline in 2017 for the CBZ therapy. At the same time, CPIC also published PGx-based dosing guidelines for CBZ inheriting *HLA* genetic polymorphisms. A recent study lead by Sukasem C et al. 2018 reported a strong association between *HLA*-*B***15*:*02* with CBZ-induced SJS/TEN/MPE [105]. Later in 2018, the universal health coverage operated by the National Health Security Office (NHSO) has declared the coverage for *HLA*-*B***15*:*02* genetic screening with a reimbursement of THB 28 per person in Thailand [169]. It takes almost 10 years from CBZ research to develop public health policy in Thailand. This approach may also be applicable to other therapeutic drug classes, e.g., for allopurinol, dapsone and cotrimoxazole etc.

#### 14.2.2. Allopurinol

Strong association of *HLA*-*B***58*:*01* allele and allopurinol induced SJS, TEN or DRESS has been reported in multiple studies conducted in Thailand [139,142,170]. A cost-effective analysis undertaken by Saokaew S et al. in 2014 revealed that pre-emptive genetic testing of *HLA*-*B***58*:*01* allele before administering allopurinol was highly cost-effective in Thailand [162]. It is worth mentioning that CPIC released PGx-based dosing guidelines of allopurinol in 2013 based on strong associations of *HLA*-*B***58*:*01* and allopurinol induced SCARs in different ethnic groups [171]. Following the robust evidence and clinical guidelines, the screening of *HLA*-*B***58*:*01* allele before allopurinol prescription is expected to be included into Thailand’s public health policy very soon. In Thailand, already the reimbursement for the *HLA*-*B***58*:*01* genetic test in susceptible risk patients expected to take allopurinol in future has been included within the universal health coverage in 2021 [117].

#### 14.2.3. Abacavir

Abacavir directed hypersensitivity reactions may usually occur in ~2–9% of patients within the first six weeks of treatment [172]. Mallal et al. 2002 first reported the evidence for the association of *HLA*-*B***5701* allele and the risk of abacavir hypersensitivity reactions in Western Australian HIV patients [173]. This association has been reported in several other ethnic groups and concluded *HLA*-*B***5701* as strong risk factor for abacavir induced hypersensitivity reactions [94,174,175,176,177]. Meanwhile, the CPIC guideline published in 2012, recommended not to use abacavir in those patients who are positive of *HLA*-*B***57*:*01* allele and also suggested *HLA*-*B***57*:*01* screening before administering abacavir and the guideline was updated in 2014 [178]. Thailand national guidelines on HIV/AIDS treatment and prevention that was originally published in 2017 but has updated in 2020, has recommended to *HLA*-*B***57*:*01* screening in these patients before initiating abacavir therapy. Appropriate intervention including cost-effectiveness and *HLA*-*B***57*:*01* screening may eliminate abacavir hypersensitivity reactions and thus PGx of abacavir may be efficiently implemented globally in HIV/AIDS clinical practices [179,180].

#### 14.2.4. Cotrimoxazole

A multicenter case-control study collecting data from Taiwan, Thailand, and Malaysia showed a strong association between *HLA*-*B***13*:*01* allele with co-trimoxazole induced SCARs (OR: 8.7, 95% CI 5.7–13.4; *p* = 7.2 × 10^−21)^ driven from SJS/TEN(OR: 2.71, 95% CI 3–5.4; *p* = 0.006) and DRESS (OR: 45, 95% CI 18.7–134; *p* = 1.1 × 10^−26^). After meta-analysis of data from Taiwan, Thailand and Malaysia and phenotype stratification indicated a strong association between *HLA*-*B***13*:*01* allele and co-trimoxazole induced DRESS (OR: 40.1, 95% CI 19.54–82.32; *p* < 0.00001 [181]. Another case-control study conducted only in Thai patients lead by Sukasem C et al. 2020 demonstrated that genetic association of cotrimoxazole driven SCARs was phenotype-specific in which *HLA*-*B***15*:*02* and *HLA*-*C***08*:*01* alleles were significantly associated with cotrimoxazole induced SJS/TEN only. In contrast, the *HLA*-*B***13*:*01* allele was significantly associated with co-trimoxazole induced DRESS only but not the SJS/TEN [182]. The association of *HLA*-*B***15*:*02* allele with cotrimoxazole induced SJS/TEN has been replicated in another study conducted in Thailand [183]. However, phenotype specific biomarkers as identified in Sukasem C et al. 2020 study for cotrimoxazole warrant further quantification in other parts of Southeast Asia for incorporation into clinical practice and making local prescribing guidelines.

#### 14.2.5. Dapsone

A very recent study identified a strong association of *HLA*-*B***13*:*01* allele with dapsone-induced SCARs O (OR: 39.0, 95% CI 7.67–198.21; *p* = 5.3447 × 10^−7^) SJS/TEN (OR: 36.0, 95% CI 3.19–405.89; *p* = 2.1657×10^−3^ and DRESS (OR: 40.5, 95% CI 6.38–257.03, *p* = 1.0784 × 10^−5^ as compared to dapsone-tolerant controls in Thai non-leprosy patients [184]. These findings are also supported by another study conducted in Thailand for non-leprosy patients [185]. Similar trends were also found with leprosy patients in other parts of Asia concluded that *HLA*-*B***13*:*01* allele was a strong predictor for dapsone induced SCARs including SJS/TEN or DRESS [186,187,188,189]. However, such genetic associations were not identified in European, Caucasian, American, African, or Oceanic population indicating that *HLA*-*B***13*:*01* allele associated dapsone induced SCARs were only prevalent in Asian population especially in China and Southeast Asia. This may be partly because either low frequency of this allele in these population (European, Caucasian, American, African or Oceanic) or may be due to not undertaking any clinical studies in these ethnic regions [188].

As strong association of *HLA*-*B***13*:*01* allele with dapsone induced SCARs has been well-established in Asian population especially in Chinese and Southeast Asian population. Genetic screening of *HLA*-*B***1301* in these population is urgently needed before dapsone therapy to optimize patient’s safety. Additionally, the genetic expertise and policy makers of these regions should focus on to create national prescribing guidelines based on the robust evidence to incorporate this into routine clinical practice. 

### 14.3. Cost-Effective Analysis

Cost-effectivity of PGx-directed treatment should be assessed before applying in clinical practice. When the routine genetic screening of pharmacogene is found to be cost-effective with the prescription drugs affected by that pharmacogene, health insurance companies are likely to influenced to reimburse routine PGx testing [190]. With the advent of cutting-edge genomic technology such as NGS and DNA microarray, the cost of genetic analysis has reduced substantially now a days and is expediting the wider acceptance of PM at the very few cost [191]. In Thailand, cost-effective analysis has undertaken for CBZ and allopurinol and found that pre-emptive screening of *HLA* was cost-effective before prescribing either CBZ or allopurinol [162,192,193]. These findings are facilitating the translation of CBZ/allopurinol PGx into the development of national drug policy and implementation in clinical practices in Thailand. 

### 14.4. Pharmacogenomics Laboratory Distribution and Local Accessibility

For ensuring high-quality genetic testing, equipment of the PGx laboratories must be high standard with international certification and staffs of these laboratories must be highly trained and qualified [194]. Patients or doctors should have easy access to the PGx laboratory for ordering PGx test. Workflow at the PGx laboratory should be smooth and efficient in genotyping and transferring the results to the patients or doctors [117]. Currently, there are 19 PGx laboratories functional in Thailand, however, PGx laboratory at Ramathibodi Hospital and Siriraj Hospital in the Mahidol University, King Chulalongkorn Memorial Hospital in the Chulalongkorn University, Songklanagarind Hospital in the Prince of Songkla University and Srinagarind Hospital in the Khon Kaen University are notables. Government of Thailand should take adequate measures to confirm the establishment of large number of PGx laboratories throughout the country to ensure easy accessibility of PGx laboratories to everyone with affordable cost for wider uptake of genetic screening at the door corner. 

### 14.5. Workflow

A systematic workflow as shown in Figure 5 is needed to implement PM efficiently in clinical service through multidisciplinary team. The PGx testing could be implemented through either pre-emptive or reactive approach. Doctors are alerted through a PGx alert software to see the patient’s PGx test report before prescribing medications in a reactive PGx testing approach. In contrast, in pre-emptive approach, patients could show the PGx test report at the first visit to doctor and doctor could then prescribe medications accordingly. Pre-emptive PGx testing approach is more useful and favorable because test reports are readily available to doctors and may expedite the entire treatment process for optimizing safety or efficacy of drugs. For successful integration of PM in daily clinical practice, following factors should take into considerations.

### 14.6. PPM CARD

The PPM card is a pharmacogenomics identity card (PGx ID) cardfrom Division of Pharmacogenomics and Personalized Medicine (PPM), Department of Pathology, Faculty of Medicine Ramathibodi Hospital, Mahidol University, Bangkok, Thailand. PPM card is a purple rectangle and wallet size card conferred to each patient having unique PGx test results in it which can carry by the patients to show their doctors anytime. Dr. Chonlaphat Sukasem first invented this low-tech plastic card at Bangkok’s Ramathibodi Hospital in 2011 and presented in front of the Global Leaders in Genomic Medicine Summit organized by the National Human Genome Research Institute (NHGRI) and National Institute of Health (NIH), USA in 2014. The scientists attended from over 20 countries in this multinational genomic medicine summit much appreciated this invention and were interested to implement this PGx ID card approach in the resource-rich countries that were discussed further extensively in the follow up meeting held in 2015 at NHGRI/NIH. The PGx ID card is a simple, rapid, and cost-effective technique which can accelerate the PM implementation in routine care setting [169]. Currently, the modified version of this simple PGx card is under-construction, in which a QR barcode (Figure 6) will be added to increase the security and confidentiality of the patient’s genetic information.

### 14.7. Multidisciplinary Team

The successful adoption of PGx in routine clinical care warranted a multidisciplinary approach where doctors, pharmacists, pathologists, medical technologists and medical informatics will be working as a team for the implementation and delivery of PGx results. Most importantly, the interaction and perception between doctors and pharmacist is key to the successful implementation of PGx in clinical settings. Doctors should be positive and reasonably receptive to the recommendations made by the pharmacists to ensure the appropriate medication provided to the patients based on the PGx test results. Doctors take final clinical decision based on recommendations made by pharmacist. Pharmacists should inform and assist doctors and patients in the use and interpretation of PGx information provided in the results [117,195]. Pathologists are usually involved in the management of PGx lab including taking and preparing samples for PGx test. Medical technologists alternatively called pharmacogenetic specialist perform lab experiment and prepare PGx test results. Medical informatics usually handle computer software and generate PGx alert based on test results.

### 14.8. Electronic Health Record

Insertion of evidence based PGx information into electronic health record (EHR) software is a crucial step to integrate the PGx alert into clinical decision support (CDS), system. The CDS system having PGx information could be used to deliver either synchronous or asynchronous interventions. In synchronous interventions, pop-up alert will appear at the time of prescribing medications advising the clinician to order a PGx screening test based on specific PGx evidence of a particular drug. In contrast, in asynchronous interventions, clinicians are notified when new PGx test results are available through either inbox message or e-mail so that they could alter or adjust the dose for the achievement of PM accordingly [196,197]. Although some countries preceded this approach, however, in Thailand, currently there is no PGx alert system integrated into the EHR. There may have serious clinical manifestations for not integrating PGx alert into the EHR system. For example, recently in Thailand, an inpatient positive for *HLA*-*B***15*:*02* allele was prescribed phenytoin but was later prescribed CBZ in the follow-up period due to lack of PGx alert system in EHR and the patient was finally died because of CBZ-induced TEN [117].

### 14.9. Counseling

Clinical pharmacogeneticians or pharmacists obtained training in PGx may provide pre or post PGx test counseling to the patients through face-to-face counseling, telephone or e-mail. The PGx counselors should provide education to the patients explaining the necessities of genotyping to facilitate CDS system. Counselors should also discuss with the patients about PGx test findings and therapeutic recommendations to make them comfortable about it [117,198].

### 14.10. Knowledge and Education

Pharmacists and doctors must take complementary roles in interpreting and implementing PGx test results in routine clinical practice [199,200]. Although many pharmacists currently have limited knowledge and understanding of PGx in general and in particular for delivering PGx test throughout the world, however, pharmacists are well-suited to offer this newly evolving service in the clinic/hospital. By supplementing their existing knowledge and expertise with available specialized training in PGx, pharmacists may provide essential support to doctors and other clinicians for making PGx testing services effective in daily clinical practice [201,202,203,204,205]. In Thailand, PGx knowledge is enhancing by the introduction of different PGx certificate program to healthcare professionals through the Pharmacy Council of Thailand (PCT) [206]. For example, three levels of pharmacists are proposed by the PCT in which Level 1 pharmacist will complete a core competency by one week training, Level 2 pharmacist will operate clinical pharmacogenomics after taking four months training and finally Level 3 pharmacist will be the clinical pharmacogenomics specialist after completing four years learning and training. In addition, the PCT has launched four months program called ‘training curriculum for certificate of proficiency in pharmacogenomics and precision medicine’ to increase expertise of PGx in the country. These initiatives must be integrated into the standard curriculum of medical disciplines and continuing medical education (CME) across Thailand for increasing PGx knowledge and confidence of the doctors for successful implementation of PGx in clinical care settings [117,207].

## 15. Expert Opinion

As there are well-established evidence for the associations of some *HLA* variants with drug induced SCARs in Thailand, therefore, it is high time for the implementation of *HLA* PGx testing in routine clinical practice throughout the country. This can be implemented either pre-emptively or reactively. Pre-emptive *HLA* PGx testing could prevent SJS/TEN substantially whereas reactive approach could facilitate the selection of right drug with right dose in right patients and may also reduce SJS/TEN significantly. As these approaches are currently functional in some parts of Thailand, the incidence of SJS/TEN has decreased dramatically since the introduction of the PGx study. However, it is highly suggested to consider the following points carefully in order to effective adherence of *HLA* testing in clinical practice.

(A)Introduce appropriate *HLA* test such as screening of family members for risk genes of CBZ, e.g., *HLA*-*B75* serotype, i.e., *HLA*-*B***15*:*02*, *HLA*-*B***15*:*08*, *HLA*-*B***15*:*11*, *HLA*-*B***15*:*21*.(B)Introduce relevant *HLA* test for all population in the settings. For example, *HLA*-*B***15*:*02* test only for Han Chinese, Thai, Indian, Malaysian and Singaporean population. In contrast, *HLA*-*A***31*:*01* test only for Caucasian, European, Japanese and Korean population.(C)Prepare alternative drugs for those patients who may be at *HLA* associated risk of SCARs and incorporated these into the local prescribing guidelines.(D)Caution should be taken for interpreting and reporting PGx test results, as some biomarkers are phenotype specific while others are universal markers as discussed in this review.

Thailand is advancing PGx research and is now working collaboratively within the country and along with some other parts of Southeast Asia to eradicate SJS/TEN completely and the associated morbidity and mortality from the country through generation of evidence-based prescribing guidelines and policy making. Thailand should now focus on ethical, legal and social issues (ELSI) more broadly, quality and cost-effectiveness of PGx testing and finally PGx education of health professionals to ensure effective and successful implementation of *HLA* PGx universally into clinical practice.

## 16. Conclusions

Associations of *HLA* genetic variants with drug induced SCARs have been extensively studied especially in Southeast Asian populations. The associations of different *HLA* alleles with the risk of drug induced SJS/TEN, DRESS and MPE are strongly supportive for clinical considerations. Prescribing guidelines generated by different national and international working groups for translation of *HLA* pharmacogenetics into clinical practice are underway and functional in many countries including Thailand. Cutting edge genomic technologies may accelerate wider adoption of *HLA* screening in routine clinical settings. There are great opportunities and several challenges as well for effective implementation of *HLA* genotyping globally in routine clinical practice for the prevention of drug induced SCARs substantially, enforcing precision medicine initiatives.

## Figures and Tables

**Figure 1 pharmaceuticals-14-01077-f001:**
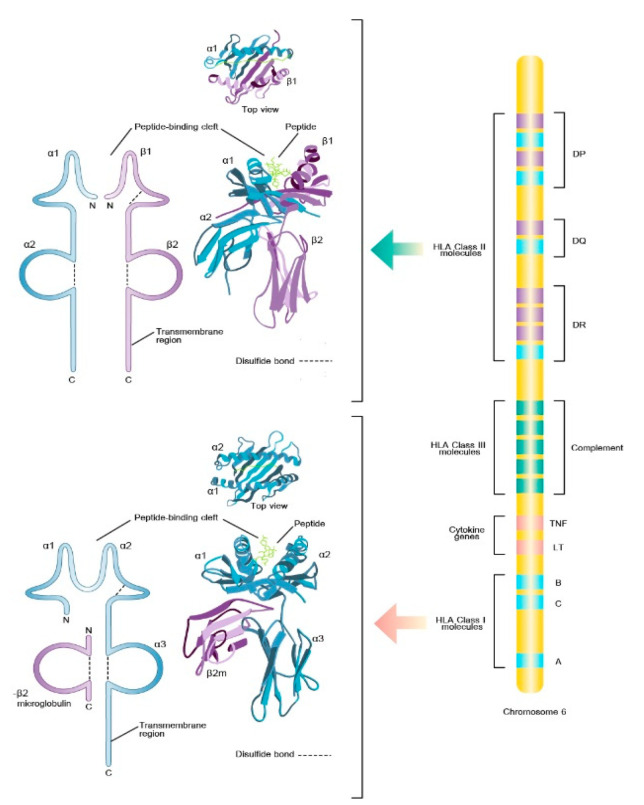
Human leukocyte antigen (*HLA*) is located on chromosome 6 and structure of *HLA* class I and class II molecules.

**Figure 2 pharmaceuticals-14-01077-f002:**
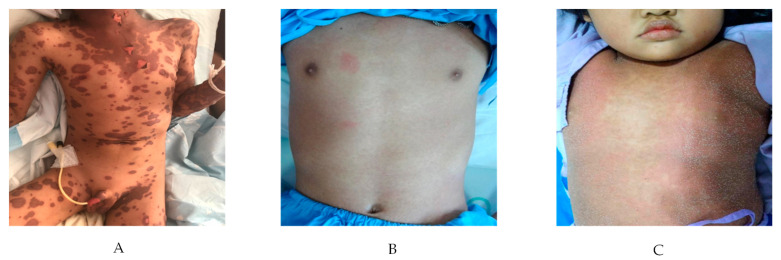
Severe cutaneous adverse drug reactions. (**A**) indicates Stevens–Johnson syndrome (SJS)/toxic Epidermal necrolysis (TEN), (**B**) indicates drug reaction with eosinophilia and systemic symptoms (DRESS) and (**C**) indicates acute generalized exanthematous pustulosis (AGEP).

**Figure 3 pharmaceuticals-14-01077-f003:**
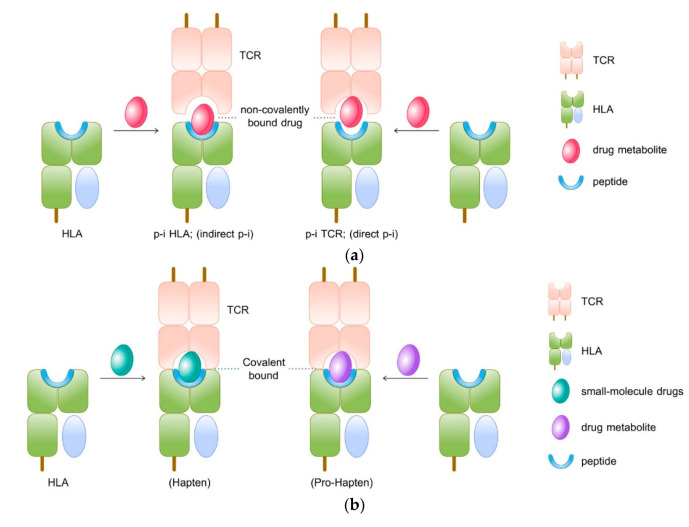
The mechanisms behind drug-induced delayed hypersensitivity reaction (DHS) are explained in three different theories (**a**) pharmacological interaction theory (p-i), (**b**) hapten/prohapten theory, and (**c**) repertoire alteration theory.

**Figure 4 pharmaceuticals-14-01077-f004:**
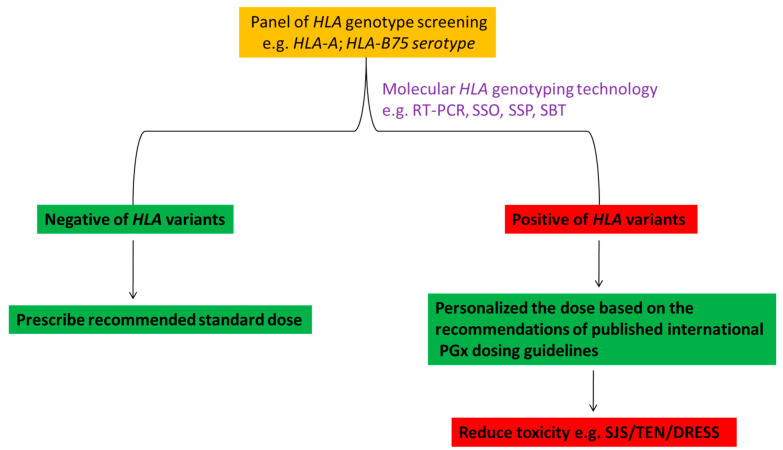
Implementation of *HLA* genotyping in routine clinical practice for optimizing safety of medications. *HLA* = human leukocyte antigen; *HLA*-*A* = *HLA*-*A***31*:*01*, *HLA*-*A***33*:*03*; *HLA*-*B75* = *HLA*-*B***15*:*02*, *HLA*-*B***15*:*08*, *HLA*-*B***15*:*11*, *HLA*-*B***15*:*21*; RT-PCR = real-time polymerase chain reaction; SSO = sequence specific oligonucleotide; SSP = sequence-specific primers; SBT = sequence-based typing; PGx = pharmacogenomics; SJS = Stevens–Johnson syndrome; TEN = toxic epidermal necrolysis; DRESS = drug reaction with eosinophilia and systemic symptoms.

**Figure 5 pharmaceuticals-14-01077-f005:**
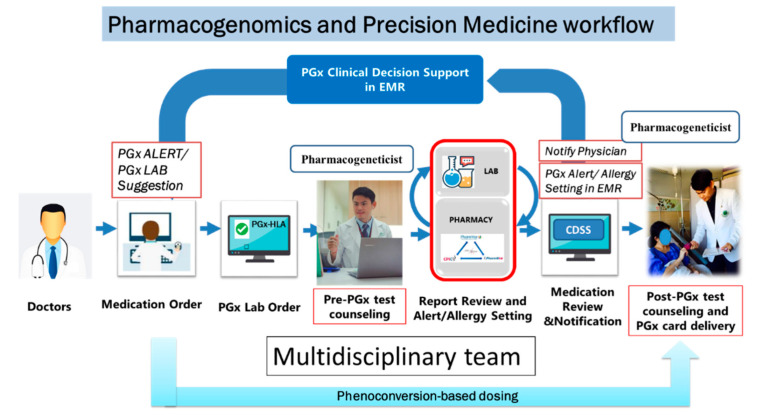
A systematic clinical workflow for implementation of *HLA* pharmacogenomics into routine clinical practice. *HLA* = Human leukocyte antigen, PGx = Pharmacogenomics; EMR-Electronic Medical Record, CDSS = Clinical Decision Support System.

**Figure 6 pharmaceuticals-14-01077-f006:**
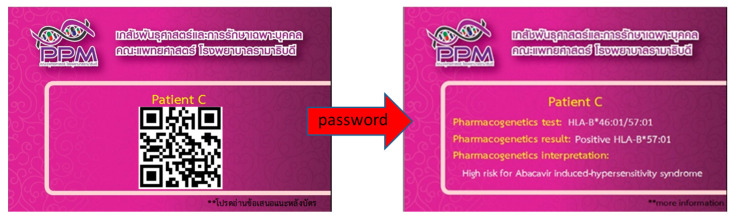
The PPM card is a pharmacogenomics identity card (PGx ID card). The QR code will be used to increase the security and confidentiality of the patient’s genetic information. It can be accessed by the pharmacogenetics information of individual patient by private password.

**Table 1 pharmaceuticals-14-01077-t001:** Comparison of the common molecular *HLA* genotyping techniques.

Method	Advantage	Disadvantage	Medical Applications
rSSO	- Intermediate to High resolution - High throughput - Robust - Large volume typing	- Allele ambiguities	- Transfusion - Transplantation - Disease associations - Pharmacogenomics
SSP	- Intermediate to High resolution - Rapid - Small volume typing	- Lower throughput - Large number of reactions - Required many thermocyclers	- Transfusion - Transplantation - Disease associations - Pharmacogenomics
Real-time PCR	- Low to High resolution - Rapid - Low volume typing	- Results reported only positive or negative for specific alleles	- Disease associations - Pharmacogenomics
SBT	- Highest resolution - Full allele - Direct identification of new alleles	- Lowest throughput - More technically demanding - Expensive - Sequencing ambiguities caused by *cis*-/*trans*-polymorphism	- Transfusion - Transplantation - Disease associations - Pharmacogenomics - Resolution of ambiguous results from other method

**Table 2 pharmaceuticals-14-01077-t002:** Diagnostic criteria for drug reactions with eosinophilia and systemic symptoms (DRESS).

RegiSCAR	Bocquet et al.	J-SCAR.
Criteria: Diagnosis when all 3 items +3/4 of all of the following clinical signs	Criteria: All 3 is required, 1 of each clinical signs	Criteria: Typical—all 7 clinical signs/criteria Atypical—5 first clinical signs
1. Acute skin eruption 2. Reaction suspected to drug-related 3. Hospitalization	1. Cutaneous drug eruption	1. Maculopapular rash develop > 3 weeks after starting offending drug 2. Prolonged clinical symptoms 2 weeks after discontinuation of the causative drug 3. Fever > 38 °C 4. ALT > 100 U/L or other organ involvement 5. Lymphocyte abnormalities ≤ 1 present -Leukocytosis > 11 × 10^9^/L -Atypical lymphocytes > 5% -Eosinophilia > 1.5 × 10^9^/L 6. Lymphadenopathy 7. HHV-6 reactivation
1. Fever > 38 °C 2. Enlarged lymph nodes ≤ 2 sites 3. Involvement ≤ 1 internal organ 4. Blood count abnormalities -Lymphocytes above or below normal limit -Eosinophils above normal limit -Platelets under normal limit	2. Hematologic abnormalities -Eosinophil > 1.5 × 10^9^/L -Atypical lymphocytes	
	3. Systemic involvement -Lymphadenopathy ≤ 2 cm -Hepatitis: transaminase ≤ 2X -Interstitial nephritis -Interstitial pneumonitis -Carditis	

**Table 3 pharmaceuticals-14-01077-t003:** The frequencies of Class I *HLA*-associated drug hypersensitivity and related drug adverse reactions.

HLA Pharmacogenetics Marker	Allele Frequency (%)						Drug	ADR Type
	Thai Population (*n* = 470) [25]	African Americans (*n* = 252) [26]	North American (*n* = 187) [26]	Caucasians (*n* = 265) [26]	Hispanics (*n* = 234) [26]	Asians (*n* = 358) [26]		
*HLA*-*A***01*:*01*	2.23	5.56	7.49	15.09	5.98	1.53	Phenobarbital	SCARs, MPE [32]
*HLA*-*A***24*:*02*	11.49	2.78	22.73	6.6	12.18	18.94	Carbamazepine	SJS/TEN [33]
							Phenytoin	SJS/TEN [33]
							Lamotrigine	SJS/TEN, MPE [33]
*HLA*-*A***30*:*02*	0	4.96	1.87	0.57	3.42	0.14	Amoxicillin-Clavulanate	DILI [34]
*HLA*-*A***31*:*01*	0.85	0.79	7.75	3.21	4.91	3.06	Carbamazepine	CADRs, SJS/TEN, DRESS, MPE [35,69,70]
	0.85	0.79	7.75	3.21	4.91	3.06	Lamotrigine	SCARs [36]
*HLA*-*A***33*:*03*	11.17	3.97	0.53	0.57	1.07	11.7	Allopurinol	SJS/TEN [71,72]
							Ticlopidine	DILI [73]
*HLA*-*A***68*:*01*	0.96	2.58	5.62	3.02	2.56	0.28	Lamotrigine	SCARs [37]
*HLA*-*B***13*:*01*	5.96	0	0	0	0	3.34	Phenytoin	SCARs [38]
							Phenobarbital	DRESS [32]
							Dapsone	DRESS [39]
							Salazosulfa-pyridine	DRESS [40]
*HLA*-*B***15*:*02*	7.66	0.2	0	0	0	4.87	Carbamazepine	SJS/TEN [35,69,70]
							Oxcarbazepine	MPE, SJS [35]
							Phenytoin	SJS/TEN [35]
							Cotrimoxazole	SJS/TEN [74]
*HLA*-*B***15*:*11*	0.21	0	0	0	0	0.28	Carbamazepine	SJS/TEN [35,69]
*HLA*-*B***15*:*13*	0.96	0	0	0	0	0.28	Phenytoin	SJS/TEN, DRESS [33,75,76]
*HLA*-*B***35*:*05*	1.91	0	0	0.38	0.85	0.14	Nevirapine	SJS/TEN, DRESS [69,77]
*HLA*-*B***38*:*01*	0	0.4	1.07	2.45	1.71	0.42	Co-trimoxazole	SJS/TEN [78]
*HLA*-*B***38*:*02*	4.26	0	0	0.19	0	6.55	Oxcarbazepine	MPE [79]
							Co-trimoxazole	SJS/TEN [78]
*HLA*-*B***51*:*01*	4.26	1.2	11.23	5.66	6.2	6.69	Phenobarbital	SJS/TEN [32]
*HLA*-*B***56*:*02*	0.11	0	0	0	0	0.28	Phenytoin	DRESS [38]
*HLA*-*B***57*:*01*	1.17	2.39	2.14	4.15	1.92	0.97	Abacavir	AHS [35,69,80]
							Flucloxacillin	DILI [81]
*HLA*-*B***58*:*01*	6.38	6.37	0.8	1.13	1.07	7.38	Allopurinol	CADRs, SCARs, MPE, SJS/TEN, DRESS [35,69,71,82]
*HLA*-*B***59*:*01*	0	0	0	0	0	0.56	Methazolamide	SJS/TEN [83]
*HLA*-*C***03*:*02*	7.77	2.78	0.27	0.38	1.07	7.66	Allopurinol	SJS/TEN [71,72]
*HLA*-*C***06*:*02*	4.26	11.31	5.62	8.68	6.84	3.62	Co-trimoxazole	SJS/TEN [74]
*HLA*-*C***08*:*01*	10.32	0.2	2.41	0	1.71	11.28	Carbamazepine	SJS/TEN [33]
							Phenytoin	SJS/TEN [84]
							Allopurinol	SJS/TEN [84]
							Co-trimoxazole	SJS/TEN [74]

**Table 4 pharmaceuticals-14-01077-t004:** The frequencies of class II *HLA*-associated drug hypersensitivity and related drug adverse reactions.

HLA Pharmacogenetics Markers	Allele Frequency (%)					Drug	ADR Type
	Thai Population (*n* = 470) [25]	African Americans (*n* = 241) [85]	Caucasians (*n* = 265) [86]	Japanese (*n* = 371) [87]	Han Chinese (*n* = 358) [88]		
*HLA*-*DRB1***12*:*02*	15.32	0.4	0	1.5	13.3	Carbamazepine	SJS/TEN [33]
*HLA*-*DRB1***13*:*02*	1.38	8.5	3.4	7.7	2.8	Allopurinol	SJS/TEN [71]
*HLA*-*DRB1***15*:*01*	8.09	16	15.8	8.5	10.8	Amoxicillin- Clavulanate	DILI [34,89]
*HLA*-*DRB1***15*:*02*	14.47	0	0.8	10	5.3	Allopurinol	SJS/TEN [71]
*HLA*-*DRB1***16*:*02*	5.96	0	0	0.9	5.3	Phenytoin	SJS/TEN [84]
*HLA*-*DQB1***06*:*02*	1.49	23.2	15.8	8.2	3.8	Amoxicillin- Clavulanate	DILI [34,89]
*HLA*-*DQA1***02*:*01*	8.72	9.1	13.2	N/A	5.7	Lapatinib	DILI [90]

**Table 5 pharmaceuticals-14-01077-t005:** Comparison of *HLA* genotyping and clinical recommendations provided by different international pharmacogenetics working bodies and drug regulatory agency.

Drug	Gene	Phenotype	Clinical Recommendations	Recom. Authority	Level of Evidence	Genotyping Recommendations
Carbamazepine	*HLA*	*HLA**-B***15**:02* negative and *HLA**-A***31**:01* negative	Use standard dose as per guidelines	CPIC	1A	Strong
		*HLA**-B***15**:02* negative and *HLA**-A***31**:01* positive	If patient is CBZ-naïve and alternative agents are available, do not use CBZ	CPIC	1A	Strong
		*HLA**-B***15**:02* positive and any *HLA**-A***31**:01* genotype	If patient is CBZ-naïve, do not use CBZ	CPIC	1A	Strong
		*HLA**-B***15**:02,* *HLA**-A***31**:01* and *HLA**-B***15**:11* carriers	Choose an alternative	DPWG	4E	Essential
		*HLA**-B***15**:02* positive	Alternative medication should be used as first-line therapy.	CPNDS	+++	B-Moderate
		*HLA**-A***31**:01* positive	Alternative medication should be used as first-line therapy	CPNDS	+++	B-Moderate
		*HLA**-B***15**:02* positive	CBZ is not recommended unless the benefits clearly outweigh the risks	FDA	-	-
		*HLA**-A***31**:01* positive	Risks and benefits should be weighed before prescription of CBZ	FDA	-	-
Oxcarbazepine	*HLA* *-B*	*HLA**-B***15**:02* negative	Use OXC per standard dosing guidelines	CPIC	1A	Strong
		*HLA**-B***15**:02* positive	If patient is oxcarbazepine naïve, do not use oxcarbazepine.	CPIC	1A	Strong
		*HLA**-B***15**:02* positive	An alternative is recommended. If not possible, it is recommended to advise the patient to report any rash immediately.	DPWG	4D	Beneficial (patients of Asian, not-Japanese and not-Korean, descent)
		*HLA**-B***15**:02* positive	Patients are at higher risk of SCARs. Genotyping is not a substitute for clinical vigilance	FDA	-	-
Abacavir	*HLA* *-B*	*HLA**-B***57**:01* negative	Use abacavir per standard dosing guidelines	CPIC	1A	Strong
		*HLA**-B***57**:01* positive	Abacavir is not recommended	CPIC	1A	Strong
		*HLA**-B***57**:01* positive	Abacavir is contra-indicated.	DPWG	4E	Essential
		*HLA**-B***57**:01* positive	Do not use abacavir	FDA	-	-
Allopurinol	*HLA* *-B*	*HLA**-B***58**:01* negative	Use allopurinol per standard dosing guidelines	CPIC	1A	Strong
		*HLA**-B***58**:01* positive	Allopurinol is contraindicated.	CPIC	1A	Strong
		*HLA**-**B*****58**:**01* positive	Choose an alternative, e.g., febuxostat or to precede treatment with allopurinol tolerance induction.	DPWG	4F	-
		*HLA**-**B*****58**:**01* positive	Results in higher severe skin reactions	FDA	-	-
Phenytoin	*HLA* *-B*	*HLA**-B***15**:02* negative	Initiate therapy with recommended maintenance dose	CPIC	1A	Strong
		*HLA**-B***15**:02* positive	If patient is phenytoin naive, do not use phenytoin	CPIC	1A	Strong
		*HLA**-B***15**:02* positive	Phenytoin can induce the life-threatening cutaneous adverse events	DPWG	4E	Beneficial (patients of Asian, but not Japanese and Korean descent)
Lamotrigine	*HLA* *-B*	*HLA* *-B* **15* *:02*	Considers genotyping of patients to be beneficial for drug safety. Avoided lamotrigine if possible, even if both the incidence and the risk increase are low	DPWG	4E	Beneficial (patients of Asian but not Japanese and Korean descent)
Flucloxacillin	*HLA* *-B*	*HLA* *-B* **57* *:01*	Regularly monitor the patient’s liver function. Choose an alternative if liver enzymes and/or bilirubin levels are elevated	DPWG	4F	-
Lamotrigine	*HLA* *-B*	*HLA* *-B* **15* *:02*	Considers genotyping of patients to be beneficial for drug safety. Avoided lamotrigine if possible, even if both the incidence and the risk increase are low	DPWG	4E	Beneficial (patients of Asian but not Japanese and Korean descent)
Flucloxacillin	*HLA* *-B*	*HLA* *-B* **57* *:01*	Regularly monitor the patient’s liver function. Choose an alternative if liver enzymes and/or bilirubin levels are elevated	DPWG	4F	-

*HLA* = Human leukocyte antigen; Recom = Recommending; CBZ = Carbamazepine; CPIC = Clinical Pharmacogenetics Implementation Consortium; DPWG = Dutch Pharmacogenetics Working Group; CPNDS = Canadian Pharmacogenomics Network for Drug Safety.

## Data Availability

Not applicable.

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
