# Peer review of "A Comprehensive Review of HLA and Severe Cutaneous Adverse Drug Reactions: Implication for Clinical Pharmacogenomics and Precision Medicine"

_pharmaceuticals, 2021, doi:10.3390/ph14111077_

Round 1
Reviewer 1 Report
This is a review summarizing of the HLA and SCARs. I have some comments that I believe might help the authors in increasing the impact of this manuscript.
I have some comments that I believe might help the authors in increasing the impact of this manuscript.
1. Please check italics. e.g., page 2, line 11: “and” is not italics.
2. Please check subscripts. Change "β2 microglobulin” to " β2 microglobulin "
3. Check an abbreviation. Please consistent use of abbreviations throughout the manuscript. A few examples:
1) Page 3, line 8: Change “T-cell receptors (TCR)” to “TCR”.
2) Page 3, line 18-91: Change “the major histocompatibility complex (MHC)” to “MHC”.
3) Page 10, line 17-18: Change “human leukocyte antigens (HLAs)” to “HLAs”.
4) Page 10, line 31: Change “natural killer cells (NK cells)” to “NK cells”.
5) Page 12, 16 and 20: Change “lymphocyte transformation test (LTT)” to “LTT”.
6) Page 13, line 36: Change “Drug reaction with eosinophilic and systemic symptoms (DRESS)” to “DRESS”.
4. Check parentheses. Page 17, line 13 and 20. Figure 3 legend.
Reviewer 2 Report
The manuscript “A Comprehensive Review of HLA and Severe Cutaneous Adverse Drug Reactions: Implication for Clinical Pharmacogenomics and Precision Medicine” by Kloypan and colleagues is a review article that describes in detail the relationship between HLA alleles and Severe Cutaneous Adverse Drug Reactions (SCARs) with implications in pharmacogenomics and personalized/precision medicine. While the novelty of this article is average because of multiple existing reviews on this subject, it is unique because the authors describe real world examples with specific cases and policies implemented in Thailand, such as the concept of a PPM card. This review will potentially be a good addition to the field. However, the article is very expansive and does not appear to be focused, as certain sections have been described in a lot more detail than what is required for context. The important and interesting points that make this article unique are lost in the sea of information which should be trimmed down to make it succinct and focused on the topic at hand. The following changes are recommended before the article can be considered for publication.
- The focus of the article is the relationship between HLA allotypes and SCARs. So the actual crux of the article starts at the section 8 “Potential Mechanism of HLA-associated Drug Hypersensitivity” on line 877. Everything before that is essentially an introduction to provide the readers with a basic background and context. In the current state of the manuscript, this is way too descriptive. The section on HLA Genotyping: Methods for identification should only provide/list the different methods (Table 1 gives enough information) briefly, and all the subsections on different typing methods should be removed. This really should not exceed one page. Currently this section is about 4 pages.
- Similarly, with the different SCARs (section 7), the authors go into excruciating details on pathology, detection, treatment etc. This entire section should be no more than a couple of pages, including a table which describes the symptoms and other relevant details of each SCAR type. The way it is written now is a better fit for a medical journal than the context of the current review. The readers only need a basic idea of SCARs for understanding this article. In short, lines 314-875 should be shortened to not more than two pages.
- The authors should recognize and cite other reviews on these topics, such the ones on the relationship between HLA allotypes and SCARs such as Fan et al. (PMID: 29333460), Fricke-Galindo et al. (PMID: 28315856), Kaniwa and Saito (PMID: 23635947), Yang et al. (PMID: 34336873), and detailed reviews of SCARs, including the etiology, pathology, treatment etc. (which should be removed from this current manuscript and referenced instead) such as Chung et al. (PMID: 27154258), Verma et al. (PMID: 24600147), Martin and Li (PMID: 18701033) etc. The authors can choose to add more to this list but these should definitely be cited.
- From section 8 onwards, the article is well written, providing new and relevant information, and need not be amended except for language (see point 5 below).
- The manuscript has numerous mistakes and needs editing for English language. The editing can be done by a native English speaker or a professional English language editing service.
Round 2
Reviewer 2 Report
The authors have satisfactorily addressed my concerns.